# Role of Leptin and Adiponectin in Carcinogenesis

**DOI:** 10.3390/cancers15174250

**Published:** 2023-08-24

**Authors:** Agnes Bocian-Jastrzębska, Anna Malczewska-Herman, Beata Kos-Kudła

**Affiliations:** Department of Endocrinology and Neuroendocrine Tumors, Department of Pathophysiology and Endocrinogy, Medical University of Silesia, 40-514 Katowice, Poland; anna.malczewska@sum.edu.pl (A.M.-H.); bkoskudla@sum.edu.pl (B.K.-K.)

**Keywords:** leptin, adiponectin, tumor microenvironment, epithelial–mesenchymal transition, angiogenesis

## Abstract

**Simple Summary:**

The involvement of leptin and adiponectin with receptors in the formation of many types of cancer as well as their impact on the clinical course of cancer patients are well established; however, the mechanisms of action of these adipokines are difficult to understand and thus need to be clarified. This review comprehensively presents in a systematized manner the implication of leptin and adiponectin in different stages of cancer development, focusing on interactions with the tumor microenvironment and its components in addition to their impact on the epithelial–mesenchymal transition and angiogenesis. A solid insight into these mechanisms is essential for the future potential use of these adipokines in cancer diagnostics and therapeutics.

**Abstract:**

Hormones produced by adipocytes, leptin and adiponectin, are associated with the process of carcinogenesis. Both of these adipokines have well-proven oncologic potential and can affect many aspects of tumorigenesis, from initiation and primary tumor growth to metastatic progression. Involvement in the formation of cancer includes interactions with the tumor microenvironment and its components, such as tumor-associated macrophages, cancer-associated fibroblasts, extracellular matrix and matrix metalloproteinases. Furthermore, these adipokines participate in the epithelial–mesenchymal transition and connect to angiogenesis, which is critical for cancer invasiveness and cancer cell migration. In addition, an enormous amount of evidence has demonstrated that altered concentrations of these adipocyte-derived hormones and the expression of their receptors in tumors are associated with poor prognosis in various types of cancer. Therefore, leptin and adiponectin dysfunction play a prominent role in cancer and impact tumor invasion and metastasis in different ways. This review clearly and comprehensively summarizes the recent findings and presents the role of leptin and adiponectin in cancer initiation, promotion and progression, focusing on associations with the tumor microenvironment and its components as well as roles in the epithelial–mesenchymal transition and angiogenesis.

## 1. Introduction

The process of carcinogenesis is complex. There are many factors affecting its initiation, promotion and progression. One of the factors involved in the formation of cancer, especially obesity-related cancers, is adipose tissue, which is currently considered to be a specific endocrine organ that produces a number of active substances—adipokines presenting autocrine, paracrine and endocrine activity.

Under normal conditions, adipokines are produced in balanced proportions. During obesity, which is an irregular state of chronic low-grade systemic inflammation [1], the production and secretion of the most well-known adipokines, leptin and adiponectin, are disrupted [2]. In obesity, excessive adipose tissue correlates with an increase in leptin levels and a decrease in adiponectin levels, while leptin gains pro-inflammatory properties. Altered levels of leptin and adiponectin, an imbalance in their production, impaired assembly, secretion and signal transduction are crucial factors for the development of cancer and induce a variety of changes leading to carcinogenesis [1,3]. The association of leptin and adiponectin with many types of cancer is well documented and includes both adenocarcinomas and squamous cell carcinomas [4,5,6]. In the context of neoplastic cells, both of these adipokines have oncologic potential and show opposing effects in tumorigenesis, activating and sharing common intracellular signaling pathways that contribute to cancer progression [7].

Studies show associations between increased leptin serum levels and increased tumor growth, whereas adiponectin exhibits an inverse and negative correlation with cancer development [5,8]. These two are implicated in different stages of cancer. Leptin is involved directly and indirectly in all the developmental stages of tumors, including initiation, promotion and metastatic progression [3,9,10,11]. It promotes proliferation, migration, survival and invasiveness of cancer cells, as well as inhibiting tumor cell apoptosis [12,13]. Anti-inflammatory adiponectin inhibits cellular metabolism, suppresses growth and tumor formation, inhibits the cell regulatory cycle, stimulates apoptosis [6], may affect cancer retardation through its insulin-sensitizing effects, and through interactions with other hormones (such as leptin, estrogen and insulin) may achieve an antitumor effect [6,7]. An important aspect is that adiponectin can antagonize the actions of leptin [8]. However, several studies suggest that adiponectin does not initiate tumor formation but may promote tumor progression. The impact of adiponectin on cancer development is twofold and may depend on the characteristics of individual tumors [3,8,14].

Leptin and adiponectin interact with the tumor microenvironment (TME) [3,15]. Crosstalk between adipokines and TME components leads to the epithelial–mesenchymal transition (EMT) activation program, which is critical for tumor malignancy and metastasis [12].

In this review, recent studies on the involvement of leptin and adiponectin in carcinogenesis are presented. The aim is to clarify the various mechanisms that link leptin, adiponectin and carcinogenesis for the future potential use of these adipokines in cancer diagnostics and therapeutics.

## 2. Characteristic of Leptin

Leptin was the first described adipokine discovered by Dr Jeffrey Friedman and colleagues in 1994 [16]. The gene for the leptin-obesity gene (ob) is located on chromosome 7q32.1. This non-glycosylated hormone is produced predominantly by subcutaneous white adipocytes as a 16 kDa propeptide consisting of 167 amino acids [6]. The protein belongs to the class I cytokines family with long four-helix motifs in the structure [1]. This peptide hormone is secreted from adipose tissue in proportion to the fat mass and the size of adipocytes. The larger the adipocytes, the more intensive the production of leptin [17]. Circulating leptin levels in healthy subjects with a normal body weight are 5–15 ng/mL, while in obese individuals, these levels can reach 100 ng/mL and exceed 250 ng/mL [6,18]. Leptin is mainly expressed in adipose tissue, but it has also been found in other non-adipose organs like the brain, muscles and gastrointestinal system [19]. Blood leptin concentrations show moderate circadian rhythms, with the lowest in the morning as well as early evening, with increases at night [20]. In the bloodstream, this adipokine occurs in free form or bound to its soluble receptor. The ratio of free to bound form shows individual variability depending on the content of white adipose tissue in the body: in obese people, it is present in the free form, while in lean people, it is found in the protein-bound form [17]. Serum leptin levels are higher in women than in men, even when adjusted for age and body mass index (BMI) [20]. This is related to the greater amount of subcutaneous fat, which secretes leptin more intensively relative to visceral fat in women [21], and the influence of sex hormones: estrogens stimulate increased leptin release from adipocytes, while testosterone shows an inhibitory effect [18].

### 2.1. Roles

The action of leptin is pleiotropic. As a “the hunger hormone” plays a critical role in the regulation of energy balance by inhibiting food intake and stimulating energy expenditure. The satiety effect, after active transport across the blood–brain barrier (BBB), is obtained in the arcuate nucleus of the hypothalamus—“the satiety center” by increasing anorexigenic peptides: proopiomelanocortin (POMC)/cocaine and amphetamine-regulated transcript (CART) and decreasing orexigenic peptide synthesis such as neuropeptide Y (NPY)/agouti-related peptide (AgRP), subsequently resulting in reduced appetite [6,22]. Besides controlling feeding, leptin modulates glucose and fat metabolism and participates in the regulation of endocrine system functioning [20]. In addition, leptin is crucial to proper reproductive function [23] and regulates the immune system [20,24,25]. Moreover, this adipokine plays a role in hematopoiesis [26] and osteogenesis [20,27].

### 2.2. Receptors

Leptin exerts its various functions at the cellular level through leptin-specific transmembrane receptors (ObRs) widespread in target organs encoded by the LEPR gene localized on chromosome 1p31.3 and activating several signaling pathways. ObRs exist in six isoforms (ObRa–ObRf) through alternative mRNA splicing. Based on structural differences, receptor isoforms are divided into three classes: long (ObRb), short (ObRa, ObRc, ObRd and ObRf), and secretory (ObRe) isoforms, all with the identical N-terminal extracellular domain that binds to leptin and different C-terminal intracellular domains [28,29]. Due to its ability to activate different cellular pathways depending on the isoform of the receptor and the interactions between them, leptin can exert diverse biological consequences. Self-regulation of ObRs as well as ligand-dependent activity are involved in the pathogenesis of leptin resistance [8].

Short forms are involved in leptin transportation and clearance [29]. ObRa and ObRc participate in transporting leptin across the BBB. ObRe is the extracellular cleaved part of ObRb and the main circulating leptin-carrier protein, which can regulate serum leptin concentration by inhibiting surface binding and endocytosis of leptin [20,28]. ObRb possesses a long, multidomain extracellular region, a transmembrane region and an elongated intracellular domain that couples to downstream signaling cascades [30]. ObRb, as the only one showing the ability to transmit intracellular signals, is expressed on immune cells and throughout the central nervous system (CNS), mainly in the hypothalamus, conditioning leptin’s participation in energy balance [20].

### 2.3. Signaling Pathways

The main and most well-established pathway linked to leptin signaling is the Janus kinase/signal transducer and activator of transcription (JAK/STAT) pathway. As leptin binds to its receptors at the cell surface, in either the CNS or periphery, it results in its dimerization, and the signal is transduced intracellularly through Janus kinase 2 (JAK2), which phosphorylates three tyrosine residues in the cytoplasmic domain of the receptor. The phosphorylated tyrosine residues on JAK and ObRb engage with downstream signaling pathways by recruiting proteins containing SH2 phosphotyrosine recognition domains, including the transcription factor signal transducer and activator of transcription 3 (STAT3). Once STAT3 is activated by phosphorylation, it is able to translocate itself to the nucleus as a dimer, where it can regulate the expression of target genes [18,31]. In addition to STAT3, leptin activation of its receptor also leads to the phosphorylation and activation of signal transducer and activator of transcription 5 (STAT5) [30]. Similar to other cytokine receptors, ObRb signaling is elicited by its associated JAK2 tyrosine kinase because it does not have intrinsic kinase activity [30].

The JAK2/STAT3 pathway is negatively regulated by many intracellular proteins. Suppressors of cytokine signaling 3 (SOCS3), an inhibitor of STAT3, regulate protein activity and promote recruitment of protein tyrosine phosphatases (PTPs) and the Src Homology 2 domain (SHP2), inhibiting JAK2 [8,18]. Via JAK2/signal transducer, other multiple downstream pathways are stimulated, like mitogen activated protein kinase (MAPK)/extracellular-signal-regulated kinase (ERK) ½ and phosphatidylinositol 3-kinase (PI3K)/protein kinase B (AKT) pathways, which regulate gene expression, cell growth and inflammation [6,10].

Another important pathway in relaying leptin signaling is the mammalian target of rapamycin (mTOR) and the calcium/calmodulin-dependent protein kinase (CaMKK2)/5′-AMP-activated protein kinase (AMPK)/acetyl-CoA carboxylase (ACC) pathways, with significant crosstalk between them [29,30].

Activation of cellular pathways also occurs due to the binding of leptin to short leptin isoforms. As a consequence of having a shorter cytoplasmic domain lacking the catalytic sites necessary for activation of JAK2/STAT3, they cannot activate this signaling pathway, but Ob-Ra, Ob-Rc, Ob-Rd and Ob-Rf are capable of binding JAK and activating other signal transduction cascades like insulin receptor substrates (IRS), initiating activation of the PI3K/AKT pathway. ObRe acts to sequester and block leptin-induced STAT3 activation [8]. A schematic summarizing leptin-induced signaling pathways is provided in Figure 1.

## 3. Characteristic of Adiponectin

Adiponectin was first discovered and characterized as an adipokine by Philipp Scherer in 1995 [32]. This member of the complement 1q family is a 244-amino acid protein that is the product of a gene located on chromosome 3q27. This adipokine consists of four domains: an N-terminal signal peptide, a variable domain (species-specific), a collagen-like domain and a C-terminal globular domain. The C-terminal globular domain simplifies binding to the adiponectin receptor and is homologous to the globular domains of collagens VIII and X, the complement factor C1q family, and the tumor necrosis factor (TNF) superfamily of proteins [32,33]. The N-terminal region of adiponectin is structurally individual and takes part in the multimerization and secretion of this adipokine [34]. Adiponectin circulates in plasma in four different forms: trimeric—low molecular weight (LMW), hexameric—medium molecular weight (MMW), multimers—high-molecular weight (HMW) and as a globular adiponectin containing only the C-terminal domain, being a product of proteolytic cleavage of adiponectin [35,36]. MMW and HMW constitute the majority of circulating adiponectin, whereas LMW is usually not detected in the bloodstream but occurs at very low concentrations in human plasma [37]. 

Adiponectin is mainly secreted by white adipose tissue (WAT) adipocytes, with the main source of production in the form of hormonally active visceral adipose tissue [21]. It is found in plasma at µg/mL levels, compared with ng/mL of leptin [33]. Under physiological conditions, this adipokine is an abundant protein in the bloodstream, accounting for about 0.01% of total serum protein and having a concentration range of 5–50 µg/mL. Adiponectin serum concentration is inversely related to BMI and insulin resistance. During pathological, metabolic disease-induced chronic inflammation circumstances, adiponectin levels decrease [38]. Levels in women are slightly higher than in men [39,40]. Androgens have an inhibitory effect on adiponectin gene expression, hence, sex-related differences in circulating blood levels are observed [41].

### 3.1. Roles

Adiponectin and its receptors are located in many tissues; therefore, it has a pleiotropic effect. This well-known homeostatic factor is involved in a variety of biological processes and biochemical events, such as carbohydrate and lipid metabolism, energy regulation, inflammation and insulin sensitivity [42,43]. The main role of adiponectin is to maintain the homeostasis of carbohydrate and lipid metabolism. In liver and muscle tissue, this hormone via AMPK, together with peroxisome proliferator-activated receptor alpha (PPARα) activation and enhancement of IRS signaling, mediates the insulin sensitizing effect, reduces the expression of gluconeogenic enzymes, inhibits gluconeogenesis and enhances fatty acid oxidation [1,19,37].

The specific biological activity of adiponectin depends strictly on its structure. Non-HMW adiponectin (i.e., complexes with lower molecular weight) presents stronger anti-inflammatory effects, while the HMW form of adiponectin, whose active form constitutes nearly 70% of circulating adiponectin in healthy subjects, has the most noticeable role in improving insulin sensitivity and protecting against diabetes, resulting in different metabolic diseases in the case of impaired multimerization sensitivity [44,45]. In the cardiovascular system, adiponectin has an anti-atherosclerotic effect [46]. In addition, this adipokine, with its anti-inflammatory and pro-inflammatory properties [37], has been shown to be a regulator of immune system cell function [25,47,48] and has an important role in human reproduction [49,50].

### 3.2. Receptors

Adiponectin’s actions are mediated through classical receptors (AdipoRs) belonging to the seven transmembrane domains receptor family, such as adiponectin receptor 1 (AdipoR1, chromosome 1q) and adiponectin receptor 2 (AdipoR2, chromosome 12p), as well as a non-classical receptor, T-cadherin [51]. Similar to G protein-coupled receptors, AdipoRs are composed of an intracellular NH2-terminal domain and an extracellular COOH-terminal domain with seven transmembrane domains. AdipoR1 is found in many tissues, such as the spleen, lung, heart, kidney and liver, being particularly abundant in skeletal muscle, whereas AdipoR2 is predominantly expressed in the liver but also in the heart, lung, skeletal muscle and kidney [32,52]. Biological effects are also dependent on specific tissues, with liver AdipoR1 involved in activating AMPK and AdipoR2 involved in activating PPARα, leading to increased insulin sensitivity [53]. Receptors differ in their affinity for particular forms of adiponectin. AdipoR1 demonstrates a higher affinity for the globular protein than the full-length adiponectin molecule; AdipoR2 has a similar affinity for both forms, whereas MMW and HMW bind T-cadherin receptors [38].

### 3.3. Signaling Pathways

Adiponectin effects are mediated via several intracellular signaling pathways. The main downstream of AdipoRs signaling is AMPK, which is crucial for adiponectin action in the liver, muscle and other organs [19]. Both the globular and full-length adiponectin molecules can stimulate AMPK activity in skeletal muscle, while only the full-length adiponectin does so in the liver [33]. As mentioned above, AdipoR1 acts principally through AMPK pathways, whereas AdipoR2 acts through the activation of the PPARα pathway [44]. Besides AMPK and PPARα pathways, interactions of adiponectin with its cognate receptors initiate the activation of several other downstream intracellular signaling cascades through p38MAPK, PI3K/AKT, mTOR, STAT3, nuclear factor-κB (NF-κB) and c-Jun N-terminal kinase (JNK) pathways [54]. The effects of adiponectin appear to be dependent on receptor-mediated increases in ceramidase activity, resulting in decreased intracellular ceramide concentrations, whereby AdipoRs themselves possess ceramidase activity [55].

Adaptor proteins containing pleckstrin homology domain (APPL) 1 protein, adaptor proteins containing pleckstrin homology domain (APPL) 2 protein, Ca^2+^, sirtuin 1 (SIRT1) and sphingosine-1-phosphate (S1P) are other emerging downstream effectors of the AdipoRs [19,32,35,37]. Adiponectin regulates the metabolic effects of insulin mainly by PI3K-AKT signaling, whose activation causes glycogen synthesis and an increase in glucose uptake but inhibits lipolysis. Insulin sensitivity increases after triggering insulin receptor substrate 1/2 (IRS1/2) by adiponectin. Additionally, adiponectin presents a cytoprotective effect as activation of AMPK suppresses mTOR and IκB kinase (IKK)-NF-κB-phosphatase and tension homologue (PTEN) signaling [37]. Furthermore, in endothelial cells, this adipokine activates protein kinase A (PKA) signaling, which promotes nitric oxide (NO) production and suppresses reactive oxygen species (ROS) generation and NF-κB signaling [56]. A schematic summarizing adiponectin-induced signaling pathways is provided in Figure 2.

## 4. Tumor Microenvironment

TME is a complex of a variety of cells and molecules surrounding tumors. This system includes tumor endothelial cells, tumor stromal cells with cancer-associated fibroblasts (CAFs) and cancer-associated adipocytes (CAAs), normal epithelial cells, immune cells with tumor-associated macrophages (TAMs), signaling molecules, blood vessels and a non-cellular part called the extracellular matrix (ECM). Closely related components and tumors constantly interact and affect each other, taking part in the growth of cancer cells and tumor progression [57,58]. Tumor factors can regulate the expression of adipokines with oncologic potential, such as leptin and adiponectin, which interact with cancer cells through TME as well [59,60]. Figure 3 illustrates the composition of TME.

### 4.1. TME

#### 4.1.1. Leptin and TME

Leptin is associated with the cellular and molecular parts of TME and can be affected through direct and indirect mechanisms that could lead to tumor cell invasion and distant metastasis [4,61,62]. Leptin treatment directly affects pro-inflammatory, angiogenic and fibrotic factors in TME [63]. The concentration of leptin is higher in plasma samples from TME blood than in plasma from peripheral blood samples of obese patients with estrogen receptor-positive breast cancer. With overexpression of the leptin gene in breast cancer tissue [64,65].

#### 4.1.2. Adiponectin and TME

Adiponectin is the most abundant adipokine in TME. The role of adiponectin in TME is not yet fully understood. There is an inverse correlation between circulating adiponectin and the various tumor antioxidant markers [66]. CAAs are a noted cause of decreased adiponectin secretion in humans [65,67]. In colorectal cancer, adiponectin modulates the inflammatory responses and influences the TME, which eventually defines the destiny of tumors [68]. This adipokine, together with the n-6 or n-3 polyunsaturated fatty acids (PUFAs) produced by periprostatic adipose tissue in prostatic cancer, has anti-tumoral effects [69]. Lower expression of adiponectin, AdipoR2, leptin and ObRs in the breast TME might be indicators of more aggressive breast cancer phenotypes [65,70,71]. The interactions of adipokines with TME are summarized in Table 1.

### 4.2. CAFs

CAFs play a pro-tumorigenic role by secreting various growth factors, cytokines and chemokines, as well as by degrading the ECM [96]. Adipocytes may be one of the origins of CAFs. The adipocytic phenotype may convert to CAFs by co-culturing with cancer cells and promoting their malignancy [97].

#### 4.2.1. Leptin and CAFs

In breast malignancy, leptin is secreted by CAFs, and it is responsible for the bidirectional interactions between CAFs and breast cancer cells, leading to the proliferation, migration and invasiveness of breast cancer cells [9,65,72]. Leptin produced by CAFs also leads to increased malignancy in non-small-cell lung cancer (NSCLC) cells via MAPK/ERK1/2 and PI3K/AKT signaling pathways in a paracrine manner [73]. Moreover, CAFs are the most abundant cells in the stroma of pancreatic tumors, so it is possible that, in the same way CAFs-secreted leptin could be involved in the invasion of pancreatic cancer cells [74].

#### 4.2.2. Adiponectin and CAFs

Adiponectin enhances tumor angiogenesis and tumor growth by inducing stromal fibroblast senescence through activation of p53 and p16-dependent pathways and by stimulating CXC chemokine ligand 1 (CXCL1) secretion from cancer cells, a key regulator of granulocyte recruitment. Thus, adiponectin deficiency could result in inhibition of tumor progression through reduction in stromal fibroblast senescence, in subcutaneous and metastasis tumor tissue, and discontinuing angiogenesis [75]. 

Carnitine palmitoyl transferase IA (CPT1A) is a rate-limiting enzyme of fatty acid oxidation (FAO), whose upregulation in CAFs promotes the proliferation, migration and invasion of colon cancer cells by increasing the ability of CAFs to release cytokines such as chemokine (C-C motif) ligand 2 (CCL2), vascular endothelial growth factor A (VEGF-A) and matrix metalloproteinase-2 (MMP-2). CAFs can induce lower CPT1A expression by reducing the secretion of adiponectin [76].

### 4.3. TAMs

TAMs represent one of the major types of immune cells infiltrating tumors. Due to their role and polarization states, two types of macrophages are distinguished: classically activated M1 macrophages and alternatively activated M2 macrophages. The M1 macrophages are implicated in the inflammatory response, elimination of pathogens and anti-tumor functions. The M2 macrophages, on the other hand, influence the anti-inflammatory response and, with their pro-tumorigenic properties, can promote the occurrence and invasion of tumor cells, leading to tumor progression [98].

Ongoing studies have shown that the TAM population is in a state of constant transition between the M1 and M2 types. The differentiation and polarization of TAMs and the proportion of each form are determined by multiple TME cytokines, chemokines, growth factors and other signals. Moreover, adipose tissue also recruits macrophages, whereas the tumor also recruits adipose stroma cells, showing a strong relationship between them. Though TAMs are able to exhibit either polarization phenotype, they closely resemble M2 macrophages as crucial modulators of TME. TAMs influence tumor progression, including cancer initiation and promotion, tumor angiogenesis, immune regulation and metastasis [99,100,101]. TAMs can also demonstrate antitumor activity. Hence, in response to microenvironmental signals, TAMs can have a dual effect on tumor growth and progression [102]. Moreover, TAMs are involved in tumor responses to therapy and modulate the efficacy of anticancer therapies such as chemotherapy, tumor irradiation, vascular-targeted therapies, targeted therapies by monoclonal antibodies and immunotherapies [103]. Nevertheless, TAMs are also considered therapeutic targets, with different types of molecular agents against TAMs as potential anti-cancer approaches [100,104].

#### 4.3.1. Leptin and TAMs

Leptin, through its connection to ObRs, which are present on the surface of inflammatory cells, regulates macrophage polarization and elevates the expression of different cytokines in TAMs. Knockdown of ObRs impacts the macrophage phenotype in TME, inhibiting breast cancer malignancy [77]. Leptin activates M2 macrophages and enhances the production of the cytokines IL-6, IL-8, IL-12, IL-18 and TNF-α, while inhibiting IL-10 and IL-4 [39,63]. Upregulated IL-18 expression, both in TAMs via activation of NF-κB/NF-κB1 and in breast cancer cells by activation of PI3K-AKT/activating transcription factor *2* (ATF-2) signaling pathways induced by leptin and IL-8 production in M2 macrophages stimulated by leptin, significantly promotes the migration and invasion of breast cancer cells. Apart from breast cancer, it is proven that IL-18 participates in the pathogenesis and metastasis of gastric cancer and melanoma [78,79]. 

Leptin might mediate the link between CAAs and M2 macrophages in metastasis [80]. Via STAT3, it promotes the polarization of M2 macrophages and enhances gallbladder cancer cell invasion and migration [81]. In colorectal cancer macrophage-specific metabolites, itaconate can exert cancer-promoting effects in M2 macrophages through downregulation of peroxisome proliferator-activated receptor gamma (PPARϒ), a cellular pathway that is regulated by leptin and acts as a tumor-suppressing factor. Also in colorectal cancer, leptin affects macrophage polarization [82].

#### 4.3.2. Adiponectin and TAMs

Adiponectin is a significant regulator of macrophage proliferation, polarization and function in inflammation. The strengthened tumor growth seen in adiponectin deficiency is likely due to the reduction in macrophage recruitment to the tumor rather than enhanced angiogenesis [83]. Undeniable adiponectin promotes the polarization of M2 macrophages [105,106,107]. It acts as an anti-inflammatory factor by suppressing M1 macrophage activation and downregulating proinflammatory cytokines via NF-kB activation, together with promoting M2 macrophage proliferation and expression of anti-inflammatory M2 macrophage markers via AMPK and AKT/PI3K-dependent mechanisms [108]. Furthermore, adiponectin regulates JmJC family histone demethylase 3 (JMJD3), which is necessary for M2 polarization, as another anti-inflammatory mechanism [109]. Adiponectin deficiency plays an important role in restraint tumor growth by reprogramming TAMs into M1 macrophages via suppressing p38 MAPK phosphorylation and partially mediating adiponectin-induced TAM polarization, which consequently limits tumor growth [14,84]. In the other direction, M1 macrophages also affect the action of adiponectin by reducing the expression of AdipoRs [37].

## 5. Matrix Metalloproteinases

Matrix metalloproteinases (MMPs) are a group of proteolytic enzymes that degrade ECM and are implicated in migration, invasion and metastasis; thereby, MMPs play important roles in cancer progression. Tissue inhibitors of matrix metalloproteinases (TIMPs) affect tumor cell invasiveness and the formation of distant metastases [110]. Therefore, MMPs have the potential to be diagnostic and prognostic markers along with therapeutic targets in cancer patients [111]. MMPs and TIMPs take part in adipogenesis [112,113,114] and adipocytes exert an important role in modifying the ECM through the secretion of MMPs such as MMP-1, MMP-7, MMP-10, MMP-11 and MMP-14 [61]. In obesity, adipogenesis undergoes dynamic remodeling, which is related to the turnover of ECM components [115]. In obese states, serum concentrations of MMP-2 and MMP-9 are elevated [116].

### 5.1. Leptin and MMPs

Many studies have reported that one of the leptin-induced cancer-cell invasion mechanisms is upregulating MMP expression. Leptin is involved in hepatocellular carcinoma development through its interaction with MMPs in the carcinogenic microenvironment [117]. This hormone promotes gastric cancer cell invasion by upregulating membrane type 1-matrix metalloproteinase (MT1-MMP) expression, and its overexpression positively correlates with clinical stage and lymph node metastasis in gastric cancer [85]. Leptin induces breast cancer progression by activating the expression of MMP-2 and MMP-9 [64,86,87]. Also, in oesophageal cancer, it stimulates the release of MMP-2 and MMP-9, increasing the invasiveness of cancer cells [88]. This adipokine may also be involved with the metastasis of gallbladder cancer as a result of increasing levels of MMP-3 and MMP-9 [89].

MMP-7 is considered to play an important role in the activation of various MMPs. High blood levels of MMP-7 are associated with the tumor progression of colorectal cancer and positively correlate with the advanced stage of ovarian cancer. Leptin increases MMP-7 expression and subsequent migration and invasion of ovarian cancer cell lines via ObRb, ERK1/2 and JNK1/2 activation signaling pathways and ObRb gene silencing suppresses leptin-induced MMP-7 expression [118]. In colon cancer, leptin-mediated expression of MMP-7 and cell invasion follow MAPK/ERK and PI3K/AKT signaling pathways. Moreover, leptin-induced MMP-7 expression activates MMP-2 and MMP-9. All of this proves leptin’s modulatory effects in the regulation of colon cancer progression [90,91].

It is also shown that leptin promotes cell invasion and migration through an increase in MMP-13 production, which serves as a downstream effector of the leptin-JAK2/STAT3 cascade responsible for cell invasion in pancreatic cancer cells. The tumoral expression of ObRb and MMP-13 correlates with lymph node metastasis [92].

Leptin may also contribute to the migration and invasion abilities of non-cancer cells, for example, endometriotic cells via the up-regulation of MMP-2 through an ObR-dependent JAK2/STAT3 signaling pathway [119] and human trophoblastic cells through MMP-14 overexpression, requiring the crosstalk between neurogenic locus notch homolog protein 1 (Notch1) and PI3K/AKT signaling pathways [120] as well as by regulating the expressions of MMP-9, TIMP1, TIMP2 and E-cadherin [121].

### 5.2. Adiponectin and MMPs

The influence of adiponectin on the expression and activity of MMPs and TIMPs is less well researched, but it plays an important role in ECM modulation [122,123,124]. Adiponectin enhances the production of MMPs such as MMP-1, MMP-2, MMP-3, MMP-9 and MMP-13 [37,125,126,127,128] and reduces TIMP1 activity—an inhibitor of MMP-9 [42,129]. Also, MMPs provide feedback on adiponectin, for example, MMP-12 induces globular adiponectin production from full-length adiponectin [130].

Adiponectin is significantly more expressed in metastatic NSCLC than in NSCLC without metastasis. In the A549 cell culture NSCLC model, transfection with adiponectin successfully increased the expression levels of MMP-1, MMP-2, MMP-9 and MMP-14, demonstrating an adiponectin-MMPs-involved mechanism in NSCLC invasion [93]. Adiponectin, via AdipoR1, inhibits mTOR through AMPK activation in renal cell carcinoma (RCC), suppresses vascular endothelial growth factor (VEGF), MMP-2 and MMP-9 and increases TIMP-1 and TIMP-2 secretion, resulting in decreased growth, dissemination and angiogenesis of RCC [94]. In liver cancer, this adipocyte-derived hormone impacts cancer growth and metastasis by downregulating gene expression levels of Rho-associated protein kinase (ROCK), IFN-inducible protein 10 (IP10), angiopoietin 1 and MMP-9 in liver tumors, as well as downregulating ROCK/IP10/angiopoietin 1/MMP-9/VEGF cell signaling in tumor tissue [95].

Furthermore, adiponectin can affect the interaction of leptin with MMPs. Via AMPK activation and through inhibition of JAK2/STAT3 (by promoting binding of SOCS-3 to ObRs and stimulating protein tyrosine phosphatase 1B (PTP1B) expression and activity—both negative regulators of this signal transduction pathway), adiponectin may block leptin-stimulated secretion of TIMP-1 and significantly stimulate MMP-1 activity [131,132]. By activating both a non-specific tyrosine phosphatase inhibitor and a specific PTP1B inhibitor, it significantly reduces the secretion of MMP-2 and MMP-9 from leptin-stimulated oesophageal cancer cells, inhibiting cancer invasion through this mechanism [88].

## 6. Epithelial–Mesenchymal Transition

Epithelial cells are organized into multicellular layers connected with each other through strong epithelial-cell junctions on both lateral sides, such as adherents junctions, tight junctions, gap junctions, and desmosomes. Another structural feature is apical–basal polarity and interaction with the underlying basement membrane. Mesenchymal cells present front-back polarity with no functional cell–cell junction components, including E-cadherin and β-catenin [133].

EMT is a cellular process whereby epithelial cells lose their characteristic polarity and cell adhesions and acquire the morphological and functional features of mesenchymal cells, which results in enhanced migratory and proliferation, apoptosis resistance, and their ability to produce ECM components. Particularly, the E-Cadherin and N-Cadherin switch and loss of E-cadherin and vimentin expression are two of the most well-defined features of EMT that can be triggered and regulated at different levels by multiple factors, including signals from TME [134,135,136]. Thus, EMT is associated with alterations of the intracellular cytoskeleton and ECM degradation, which cause local invasion and subsequent dissemination to distant tissues [137].

EMT types are specified. Type I and II EMT are associated with many physical processes: embryonic and organ development, wound healing, tissue regeneration, and fibrosis [138,139] while type III EMT is crucial for tumor malignancy and plays important roles in cancer progression [140,141]. EMT is frequently activated during metastasis and is directly linked to the acquisition of cancer stem cell (CSC) properties [142].

The most common and, at the same time, most lethal human malignancies are derived from epithelial tissues. Cancer-associated deaths are mostly caused by metastatic disease. EMT—an important phenomenon for cancer cells—is activated during either tumorigenesis or metastasis [136,143]. Further, due to the involvement of EMT in the metastatic process and the various states produced during EMT, targeting and manipulating this process provides a number of opportunities to influence cancer progression and can be used for therapeutic strategies in cancer during different procedures [143,144]. Currently, there are few clinical trials testing the therapeutic efficacy of agents specifically designed to suppress EMT program expression [141]. Regulators involved in EMT may be used as biomarkers and for therapeutic targeting [136].

### 6.1. Leptin and EMT

Leptin signaling activates multiple pathways and affects transcriptional factors that drive reprogramming of gene expression underlying epithelial loss and expression of mesenchymal features associated with loss of cell–cell junctions and apical-basal polarity [143,145]. Several studies describe that leptin promotes the expression of mesenchymal markers and decreases epithelial markers, in addition to promoting EMT-related processes such as cell migration and invasion and a poor prognosis in patients with numerous types of cancer [12,60,62,146]. 

The association between leptin and EMT has been most well studied in breast cancer. Chronic leptin treatment induces EMT in non-tumoral breast epithelial MCF10A cells, which leads to the belief that high leptin expression in normal breast tissue with the assistance of EMT contributes to a higher risk of breast cancer [147]. Indeed, leptin, through cytosolic tyrosine kinases such as steroid receptor coactivator (Src) and focal adhesion kinase (FAK) activation, promotes the expression of EMT-related transcription factors and invasion in MCF10A cells [87]. Leptin is involved in the regulation of EMT in triple-negative breast cancer (TNBC), and EMT regulators are major targets of TNBC [148,149]. In breast cancer cells (BCCs), leptin induces EMT by β-Catenin activation through AKT/glycogen synthase kinase 3 beta (GSK3β) and metastasis-associated protein 1 (MTA1)/Wnt family member 1 (Wnt1) pathways, as well as functional interactions between leptin, Wnt1 signaling components and MTA1—an important modifier of Wnt1 signaling [150]. Other research demonstrates that leptin-induced EMT in BCCs requires IL-8 activation via the PI3K/AKT signal pathway [151]. Further studies also suggest that leptin promotes EMT in BCCs via the activation of the PI3K/AKT signaling pathway but also via the overexpression and activation of pyruvate kinase M2 (PKM2) [152].

Another leptin signaling pathway for EMT is the transforming growth factor beta 1 (TGFB1) pathway, a central player in EMT that interacts with other EMT signaling pathways. Support for breast cancer invasiveness and CSC behavior by leptin is mediated through the binding of TGFB1 to its receptor. Further, antagonizing the TGFB-TGFB-receptor interaction degrades the EMT-promoting effects of leptin [142]. BCCs co-cultured with adipose stromal/stem cells isolated from obese women (obASCs) demonstrated enhanced expression of EMT and metastasis genes (SERPINE1, MMP-2, IL-6), and knockdown of leptin produced by obASCs significantly reduced tumor volume and decreased the number of metastatic lesions to the lung and liver [86].

The stromal cell-derived factor 1 (SDF-1) is a chemokine frequently produced in large amounts by target organs where metastasis occurs. Chemokine receptor type 4 (CXCR4) is the sole receptor for SDF-1, which was also recently described as a marker of EMT. Leptin induces tumor dissemination and metastasis of BCCs to bone tissue by activating the SDF-1/CXCR4 axis, and upregulation of CXCR4 contributes to bone metastasis and poor survival. Moreover, leptin downregulates expression of the epithelial marker E-cadherin and upregulates expression of the mesenchymal marker vimentin in BCCs, and inhibition of ObRs in BCCs significantly reduces the incidence of leptin-induced EMT [153].

In esophageal adenocarcinoma (EAC), leptin produced by peritumoral adipose tissue with increased cell diameter upregulates expression of EMT markers such as alpha-smooth muscle actin (α-SMA) and E-cadherin and thus may promote extension and penetration by cancer cells into neighboring tissues [154]. Snail is a zinc-finger transcriptional repressor that induces EMT and downregulates E-cadherin expression. It is a metastatic suppressor that is lost and shifts to N-cadherin, which is one of the typical features of EMT [132]. In gastric cancer, leptin increases the mRNA and protein levels of those EMT markers—Snail and N-cadherin—inducing EMT in such a manner [155].

In cholangiocarcinoma, leptin significantly stimulates EMT by provoking cell migration and invasion, impacting multiple levels of EMT promoters (reducing E-cadherin and β-catenin expression in addition to enhancing vimentin and N-cadherin expression) along with the proangiogenic capability of cholangiocarcinoma cells through the microRNA-122/PKM2 axis [156]. Leptin can regulate EMT through the activation of the Hedgehog (Hh) pathway, which induces hepatic stellate cells acquisition/maintenance of a myofibroblastic phenotype [157]. Leptin significantly increases tumor necrosis factor alpha (TNF-α) secretion through the activation of p38 and JNK/MAPK [28] and TNF-α can induce cancer invasion and metastasis associated with EMT in colorectal cancer [158], suggesting a potential effect of leptin on EMT in colorectal cancer as well.

A study on A549 human lung cancer cell lines shows that leptin can significantly enhance the expression of transforming growth factor beta (TGF-β), which is a direct inducer of EMT [159]. Moreover, leptin, through a mechanism dependent on the activation of the ERK signaling pathway, increases EMT-induced tumor phenotypes in lung cancer cells too [160].

Research in prostate cancer cells demonstrates that leptin, by stimulating the STAT3 signaling pathway, promotes EMT and migration of prostate cancer cells [161]. Leptin treatment upregulates EMT in ovarian and pancreatic cancer cell lines as well [148].

MMPs can induce EMT in two ways: directly by degrading adherents and tight junction proteins (MMP-2, MMP-9) [59,87,162] and indirectly by TGF-β and TGF-β-related protein activation (MMP-2, MMP-9, MMP-13 and MMP-14) [59,138,163]. In addition, M2 macrophages could play key roles in cancer progression, including the promotion of EMT. Thus, leptin-induced stimulation of M2 macrophages and MMPs affects EMT [4].

### 6.2. Adiponectin and EMT

Compared to leptin, less is known about the relationship between adiponectin and EMT. Nevertheless, several researchers have attempted to investigate these relationships. Evidence suggests that adiponectin inversely correlates with cancer progression, in part due to the reversal and inhibition of EMT [60,145].

The insulin-like growth factor-I receptor (IGF-IR) contributes to the establishment and maintenance of EMT as well as the development and maintenance of CSC in breast cancer. In ERα-negative BCCs, adiponectin has an antagonistic effect on IGF-IR signaling through activation of AMPK and inhibition of mTOR signaling, indirectly blocking IGF-IR-induced EMT [164]. However, AdipoR1 can regulate EMT in breast cancer as a direct target of microRNAs (miRNAs) miR-221 and miR-222 (miR-221/222) and provides an additional node by which miR-221/222 induces BCCs EMT. In breast cancer, miR-221/222 is differentially expressed in the clinically more aggressive basal-like subtype compared to the luminal subtype, and upregulation of miR-221/222 provokes EMT, which shows that AdipoR1 may play an important role in breast cancer progression and metastasis by implication [165].

In nasopharyngeal carcinoma (NPC) patients, serum adiponectin level is inversely correlated with tumor stage, recurrence and metastasis, and low serum adiponectin level correlates with poor metastasis-free survival. EMT is involved in the invasion and migration of tumor cells, and adiponectin via AdipoR1 has a reversing impact on this process through two mechanisms. Firstly, adiponectin treatment significantly increases the expression of E-cadherin and Claudin-1 while decreasing the levels of N-cadherin, MMP-2, MMP-9, Snail, Slug and vimentin, further blocking the EMT process. Secondly, recombinant adiponectin or a specific adiponectin receptor agonist (AdipoRon) mediates the inhibitory effect on activation of NF-κB and STAT3 signaling pathways, which are leptin-induced signaling pathways intimately involved in promoting EMT and play important roles in the metastasis of NPC [166].

The results of the study on NSCLC revealed that adiponectin is an important negative regulator of NSCLC migration and invasion through the reversal of the EMT process. After adiponectin administration, NSCLC cells displayed increased epithelial marker expression and downregulation of mesenchymal marker expression. Adiponectin upregulated E-cadherin and downregulated vimentin expression. What is more, AdipoR1 or AdipoR2 knockdown eliminated the inhibitory effects of adiponectin on migration and invasion in NSCLC and EMT, which proves that both AdipoRs mediate the adiponectin-associated signaling pathways to regulate EMT [167].

In colon cancer, adiponectin reduces cell migration ability and survival rate in association with the induction of oxidative stress and the regulation of cytokine expression (IL-6, IL-8 and IL-10). Nonetheless, Western blot analysis performed on E-cadherin and vimentin, two EMT-crucial markers in carcinogenesis, indicated that adiponectin does not influence EMT transition [135,168].

Adiponectin silencing in 22RV1 cells—human prostate cancer cell—downregulates the expression of epithelial markers: E-cadherin and zonula occludens-1 (ZO-1), but upregulates the expression of mesenchymal markers: zinc finger E-box binding homeobox 1 (ZEB1), vimentin and Snail. In addition, epigenetic modifications of adiponectin are involved during the EMT process. TGFB1 treatment in 22RV cells significantly decreased the expression levels of adiponectin, suggesting that adiponectin may play an inhibitory role in EMT. It shows that silencing endogenous adiponectin could promote the proliferation and invasion of prostate cancer cells via the EMT process. In consequence, in prostate cancer, adioponectin may function as a potential tumor suppressor but is commonly downregulated by DNA promoter methylation [169].

In view of the fact that adiponectin inhibits proliferation through blocking phosphorylation of GSK-3β, preventing β-catenin activation, and nuclear translocalization in breast cancer, this effect has been investigated on GSK-3β signaling pathways in RCC cells. In RCC, adiponectin administration also inhibited the phosphorylation of GSK-3β and decreased the accumulation of β-catenin. Additionally, silencing AdipoR1 restored the expression of EMT-related proteins, so activating the adiponectin AdipoR1 axis could hinder their expression. Inhibition of GSK-3β/β-catenin pathway by adiponectin was involved in the reduction in RCC cell motility and invasiveness without an antiproliferative effect, thus downregulating the phosphorylation of GSK-3β can stop EMT [170].

As mentioned earlier, the hallmark of EMT is the upregulation of N-cadherin, followed by the downregulation of E-cadherin [134]. Although T-cadherin, as a non-classical adiponectin receptor localized on the apical cell surface, like EMT epithelial and mesenchymal E-cadherin and N-cadherin, belongs to the cadherin superfamily, due to the lack of a transmembrane and cytoskeletal domain, T-cadherin does not participate in cell-cell adhesion but plays an important role in intracellular signaling. However, studies investigating the role of T-cadherin in cancer describe T-cadherin as a tumor suppressor in many cancer types, and its loss is associated with a more aggressive course of numerous cancers, which also indicates the involvement of adiponectin in carcinogenesis and the therapeutic potential of T-cadherin [171,172].

## 7. Angiogenesis and Vasculogenic Mimicry

Angiogenesis is the compound formulation of new blood vessels from pre-existing vessels. Involving protease production, endothelial cell migration and endothelial cell proliferation. Vascular tube formation, anastomosis of newly formed tubes, synthesis of a new basement membrane, incorporation of pericytes and smooth muscle cells, as well as the activation of proangiogenic and antiangiogenic factors in response to many agents and cytokines [173,174]. In vasculature, non-endothelial cells are involved, such as progenitor cells and CSC [175]. It is a physiological process observed in embryonic development and regenerative processes such as wound healing. However, angiogenesis can also play a role in diseases, including cancer.

Oxygen and nutrients are essential for staying alive, along with the proliferation of malignant cells. As a result of their small size, the delivery of these substances is conducted by diffusion. After the tumor has reached a size of more than 1–2 mm, it requires blood vessels to continue growing. Therefore, the tumor requires the ability to spread and induce the formation of a functional vasculature, having to reside close to blood vessels in order to access the blood circulation system [54,175].

Pathological neovasculature formation follows in the course of angiogenesis and epithelial cell independent vasculogenic mimicry (VM) [176,177,178]. These two mechanisms of blood supply complement each other with vasculogenesis. In cancer, angiogenesis can arise in a variety of forms, namely sprouting angiogenesis, intussusceptive microvascular growth and glomeruloid microvascular proliferation, whereas VM develop independently of normal blood vessels or angiogenesis [179]. During VM, tumor cells model vessel-like structures in the form of their own fluid-conducting channels without the involvement of endothelial cells [178,180,181]. This intratumoral microcirculation pattern, as a plasticity of aggressive cancer cells, is associated with high tumor grade, invasion, metastasis and poor clinical outcomes in patients with malignant tumors [176,177,181]. Molecules, markers and signaling pathways involved in VM belong to vascular endothelial (VE)-cadherin (also known as CD144), VEGF, tissue factor (TF), epithelial cell kinase 2 (EphA2), Wnt, cyclooxygenase-2 (COX-2), MMPs and hypoxia [181,182]. Furthermore, in carcinogenesis, the cancer cells switch their phenotype to angiogenic and themselves produce significant amounts of proangiogenic substances, including VEGF, fibroblast growth factor (FGF), platelet-derived growth factor (PDGF), IL-6 and IL-8 [39]. Tumor growth is vascularization-dependent, and blocking angiogenesis has been shown to suppress tumor growth.

### 7.1. Angiogenesis in Obesity

Impaired angiogenesis paired with inappropriate ECM remodeling and inflammation are the main contributors to the pathogenesis of dysfunctional adipose tissue, as well as perivascular adipose tissue (PVAT), which surrounds most mammalian blood vessels [183,184]. In obesity, there is vascular endothelial dysfunction, and adipose tissue presents lower expression of markers of angiogenesis. Insufficient angiogenesis, as a consequence of pathological remodeling of the ECM, leads to adipose tissue hypoxia and inflammation, which induce adipose stromal cells to release exosomes enriched in VEGF, which may cause adipose tissue rearrangement and stimulate extended angiogenesis [115,185,186]. VEGF signaling through angiogenesis plays an essential role in viability and adequate adipose function. To promote neovascularization during its expansion, adipose tissue expresses various other angiogenic growth factors besides VEGF, FGF, placental growth factor (PlGF) and leptin [187]. Excess adipose tissue leads to high demand for a vascular supply, resulting in regions with insufficient vascularization and hypoxia. Hypoxia provokes angiogenesis through the induction of proangiogenic factors: VEGF and hypoxia-inducible factor 1 alpha (HIF1α) [188], which interact with each other [40] and activate inflammatory pathways, macrophage recruitment and dysregulation of adipocytokine secretion [11].

Regulators of angiogenesis include leptin and adiponectin, which present in general opposing actions in endothelial cells and activities on the angiogenesis process [189,190]. A summary of these adipokines effects on angiogenesis is provided in Table 2. 

#### 7.1.1. Angiogenesis, VM and Leptin

Most of the literature shows that leptin has a proangiogenic effect, but there is also data showing that leptin does not regulate circulating angiogenesis-related factors [241,242]. 

Leptin, by activating different pathways, enhances the proliferation, migration and differentiation of endothelial cells, participates in blood vessel formation and promotes the expression of various angiogenic factors and molecules related to the development of VM [243,244]. It is reported that leptin has a synergistic effect with fibroblast growth factor 2 (FGF2) or VEGF in stimulating blood vessel growth, and the angiogenic effect of leptin is equivalent to the angiogenic effect of VEGF [243,245], demonstrating how strongly this adipokine affects the formation of new blood vessels. Leptin–VEGF crosstalk supports the progression of cancer disease through the migration and invasion of cancer cells. Likewise, cancer cells affect leptin and VEGF synthesis in human adipose stem cells [246].

Overall, leptin can facilitate vasculogenesis and function as a proangiogenic factor in two ways. One is directly binding to ObRs on vascular endothelial cells and vascular smooth muscle cells while activating leptin’s main intracellular signaling pathway, JAK2/STAT3, thereby enhancing the proliferation of vascular endothelial cells. Via JAK2/STAT3, leptin promotes endothelial cell differentiation and may enhance the generation of them derived from embryonic stem cells (ESCs), promoting angiogenesis in embryonic vessels in this way. The second mechanism is stimulating the production of angiogenesis factors such as VEGF, FGF and IL-6 as well as suppressing apoptosis through increasing the production of anti-apoptotic factors B-cell CLL/lymphoma 2 (Bcl-2) by which it influences the formulation of the new blood vessels [4,6,10,65,191,192,247]. Additionally, during leptin-induced activation of JAK2/STAT3, STAT3 dimers can activate the transcription of genes such as SOCS3, which modulates the effects of leptin on cells, as well as VEGF, both involved in angiogenesis [194,241].

Besides JAK2/STAT3, activation of other signaling pathways is involved in leptins pro-angiogenic properties. By activating NF-kB and ERK1/2, this adipokine causes vascular smooth muscle cell proliferation by promoting the transition from the G1 to the S phase [195]. Leptin stimulates the proangiogenic capability of cholangiocarcinoma cells through the miR-122/PKM2 axis [156]. miR-122 serves as a tumor suppressor and downregulator of vascular endothelial growth factor C (VEGFC) expression, leading to the inhibition of bladder cancer growth and angiogenesis [209]. PKM2 facilitates tumor growth and promotes tumor angiogenesis by regulating HIF-1α through NF-κB activation, which ultimately triggers VEGF-A secretion and subsequent blood vessel formation [210,248]. Via the AKT and Wnt signaling pathways, leptin is involved in endothelial cell proliferation and tube formation and in such a manner, leptin increases the expression of VEGF, kinase insert domain receptor (KDR), CD31, CD144 and PDGF—crucial players in cell migration and angiogenesis [197]. Moreover, to increase the expression of VEGF and leptin in the same way as PDGF, it requires both the activation of mTOR and the generation of ROS via NADPH oxidase. Whereas in the presence of rapamycin, a specific mTOR inhibitor, leptin and PDGF are no longer able to activate mTOR [213]. 

Studies have revealed multiple mechanisms underlying leptin’s proangiogenic features. Leptin regulates the function of circulating angiogenic cells (CACs), a progenitor cell type that is involved in angiogenesis. Leptin, through activation of Src kinase and integrin αvβ5, supports the angiogenic properties of CACs, promoting the adhesion and incorporation of CACs into structures provided by endothelial cells [198].

Low levels of oxygen are also common in cancer tissues, and the metabolic adaptation of cancer cells to hypoxia is crucial to keeping these cells alive [249]. Secondary to hypoxia in a neoplastic tumor, the expression of pro-apoptotic proteins (Bid, Bad and Bax) decreases, while on the contrary, the expression of anti-apoptotic proteins (Bcl-2, Bcl-XL) increases [39]. HIF1α, as the key factor in cell response to cellular hypoxia, stimulates inflammation and angiogenesis by increasing the synthesis of VEGF, which results in escalating tumor growth, aggressiveness and metastasis [11,250]. An additional element indicating a link between leptin and angiogenesis is the observed increase in leptin mRNA expression during increased hypoxia. The accumulation of HIF1α promotes the induction of several gene targets, such as leptin and ObRs, in adipocytes, fibroblast and tumor cells [233]. Through the promotion of angiogenesis by HIF1α, leptin may increase the invasiveness of oral squamous cell carcinoma [234], gastric cancer cells [235] and pancreatic cancer cells [236]. The expression of both HIF1α and ObRs is higher in the advanced stages of tumor development [234]. Moreover, leptin may drive cancer progression in a hypoxic environment and when mitochondrial respiration is impaired by sustaining aerobic glycolysis, which can stimulate cancer survival in an adverse metabolic microenvironment by sustaining HIF-1α activity. In vitro, treatment with leptin up-regulated HIF-1α and increased adhesion and invasion of prostate cancer cells cultured in oxygen limiting conditions [237].

Excessive ROS production leads to oxidative stress, which contributes to the pathogenesis of cancer. Through the activation of NOX enzymes, leptin acts as a potential activator of ROS production in human epithelial mammary cells [231]. In a ROS/HIF1α-dependent manner, accompanied by increased production of VEGF and IL-6, leptin participates in human umbilical vein endothelial cell (HUVEC) tube formation [232].

Leptin’s pro-angiogenic action is related to the IL-1 system. IL-1 is a noted inducer of VEGF expression in different tissues, and VEGF signaling transduction is required for IL-1 induction. Leptin induces several signaling pathways, such as JAK2/STAT3, MAPK/ERK 1/2, PI3K/AKT1, protein kinase C (PKC), p38 and JNK, to upregulate the translational and transcriptional expression of the IL-1 system in BCCs. Leptin and IL-1 signaling can activate NF-kB and increase the levels of VEGF and Bcl-2, which could be linked to breast cancer progression. In addition, leptin and IL-1 positively regulate VEGF/VEGFR2 and leptin-mediated upregulation of VEGF/VEGFR2 is partially mediated by IL-1/IL-1 type I receptor signaling [221]. Likewise, in breast cancer, Notch, IL-1 and Leptin Crosstalk Outcome (NILCO) are decisive for leptin upregulatory effects on cell proliferation and migration as well as VEGF/VEGFR-2 expression [222] but the level of NILCO biomarker expression depends on the presence of estrogen receptors [223].

TF, the integral membrane glycoprotein necessary for hemostasis, which expresses on its membrane almost all types of cancer, also participates in angiogenesis and VM. In human breast cancer MCF-7 cells, leptin upregulates TF expression and increases its activity [229].

Interestingly, leptin induces COX-2 expression and enhances the production of prostaglandin E2, as well as increasing aromatase expression in BCCs through COX-2 expression, which is correlated with COX-2 upregulation [226]. Functional endothelial p38MAPK/AKT/COX-2 signaling axis is mandatory for pro-angiogenic effects of leptin and is upregulated by ObRb-dependent activation of vascular endothelial growth factor receptor 2 (VEGFR2). Thus, VEGFR2 functions as a mediator of leptin-stimulated COX-2 expression and angiogenesis [227]. In addition, leptin takes part in the phosphorylation of VEGFR2 independently of VEGF in endothelial cells and BCCs [251].

MMPs are known to be involved in ECM remodeling and new blood vessel formation both during angiogenesis and VM, which are common mechanisms of obesity and cancer. Indeed, endothelial cells produce and secrete MMPs, and adipogenesis is dependent on angiogenesis [112]. MMPs particularly dedicated to vascularization are MMP-1, MMP-2, MMP-7, MMP-8, MMP-9 and MMP-14 [252].

MMP-1 activity stimulates upregulation of VEGFR2 and endothelial cell proliferation through stimulation of protease-activated receptor-1 (PAR-1) and activation of NF-κB [253]. MMP-2 and MMP-9 give rise to the modulation of the dynamic remodeling of the ECM and have the ability to proteolytically degrade denatured collagen in the vascular basal membranes, implying their involvement in angiogenesis [59,252]. Similarly, MMP-7, MMP-8 and MMP-14 participate in promoting and regulating neovascularization. 

VEGF stimulates the activity of these enzymes responsible for the destruction of the ECM [39], but MMP activation can be induced by several other angiogenic factors containing leptin, which increases the production of MMPs from endothelial cells [156]. Leptin indirectly augments angiogenesis through the induction of MMP-1 [85], MMP-2 and MMP-9 [64,86,87,88,239], MMP-7 [90,91] and MMP14 [120] activities and their signaling pathways are necessary for leptin angiogenesis [214]. The angiogenic effects of leptin involve promoting the mobilization of vascular progenitor cells and neovascularization by NADPH oxidase isoform 2 (NOX2)-mediated activation of MMP-9 [173]. 

TAMs and adipose tissue macrophages promote tissue remodeling and angiogenesis. M2 macrophages produce a broad range of angiogenic and growth factors, including, among others, VEGF, PDGF, transforming growth factor beta alpha (TGF-α), TGF-β, IL-1, IL-6 and MMPs. They promote the proliferation and migration of tumor cells to vascular endothelial cells, matrix degradation and ultimately angiogenesis [11,146,254]. Leptin’s paracrine actions can attract and further affect TAMs and stromal cells, which express ObRs to TME and secrete VEGF and IL-1, respectively, which in turn promote angiogenesis [255]. Leptin signals also impact other stromal cells like CAFs, causing their proliferation [255,256] and which together with their matrix proteins are essential in neoangiogenesis regulation [257,258]. EMT regulators and EMT-related factors—Twist, ZEB1, Snail and Slug/Snal2, highly regulated in VM-forming tumor cells can also make a relevant contribution to the VM-forming process [240]. 

Leptin has been linked to aberrant angiogenesis in many types of cancer. Human glioblastoma multiforme (GBM) cell lines express leptin mRNA and ObR, which can stimulate tube formation and enhance proliferation of endothelial cells, while inhibitors of ObR block these effects [193]. Thus, the relationship between leptin, ObR and VM formation was examined in this cancer. VM recognized by CD31-/PAS+ immunohistochemical staining in glioblastoma tissues positively correlates with leptin and ObR overexpression and ObR-positive glioblastoma cells associate with the glial–mesenchymal transition, a process implicated in VM. High VM or ObR expression, presents a poorer prognosis for overall survival times [259]. In esophageal adenocarcinoma, increased adipocyte diameter in the peritumoral adipose tissue expresses higher levels of leptin and is also associated with increased levels of CD31 [154]. In endometrial cancer cell lines, leptin promotes cell proliferation and invasion through the JAK2/STAT3, PI3K, ERK2 and COX2 pathways and regulates the expression of pro-angiogenic factors such as VEGF, IL-1β, HIF-1, leukemia inhibitory factor (LIF) and their respective receptors through the activation of JAK2-PI3K-ERK-mTOR [12,118]. Leptin is able to increase VEGF secretion in MCF7 and MDA-MB-231 BCCs. Exposed HUVEC to leptin not only results in increased proliferation and migration of HUVEC but also in the formation of more elongated and bifurcating capillary-like tubes [190]. VEGF is stimulated by HIF-1α and NFκB [6] and leptin, through the activation of HIF-1α and NF-kB via canonic (MAPK, PI-3K) and non-canonical (JNK, p38 MAP, PKC) signaling pathways, upregulates VEGF in breast cancer [196]. Also, leptin induces VEGF synthesis and the formation of new blood vessels through the activation of the PI3K/AKT/mTOR/S6 kinase signaling pathway [211] and by activating the ObR and MAPK pathways (p38, ERK and JNK), resulting in strengthening the activating protein-1 (AP-1) transcription factor for VEGF promoter binding [212].

#### 7.1.2. Angiogenesis, VM and Adiponectin

Undeniably, adiponectin exerts significant effects on vessel angiogenesis. However, there are discrepancies in the literature regarding the role of adiponectin in angiogenesis, and conflicting evidence has been published [36,54,214]. Adiponectin may present pro-angiogenic or anti-angiogenic function that depends on cancer cells manipulation to their own advantage to grow, proliferate or evade immune surveillance [68], which suggests that it has a complex influence on cancer cells that depends on the tumor environment and cell type [224]. Nevertheless, a possible explanation for the observed contradictory results in research may be differences in the cell types used and in the microenvironments between in vivo and in vitro studies [54].

Adiponectin-mediated signals in endothelial cells implicate AdipoRs and the activation of pathways such as AMPK, AKT, rous sarcoma kinase (RAS)-ERK1/2, MAPK and endothelial nitric oxide synthase (eNOS)/NO [189,203,204].

T-cadherin is required for adiponectin-induced migration and proliferation of endothelial cells. Expression of T-cadherin is critical for the revascularization actions of adiponectin, and the T-cadherin/adiponectin interaction is substantial for vascular homeostasis [199,214].

As a proangiogenic factor, adiponectin stimulates angiogenesis [200] by promoting endothelial cell migration and proliferation [199] as well as by protecting some endothelial progenitor cell (EPC) subpopulations against apoptosis and therefore modulating EPC functions [201]. Adiponectin supports angiogenesis in in vitro cell culture systems and other angiogenesis models by activating the AMPK and PI3K/AKT signaling pathways [75]. AdipoRon administration reduces endothelial cell viability and promotes angiogenesis and migration capacity. The effects induced by AdipoRon administration are accompanied by an increase in the expression of the main endothelial angiogenic factors: CXCL1, VEGF-A, MMP-2 and MMP-9 [214]. Preconditioning mesenchymal stem cells with AdipoRon increases cell survival, migration and angiogenesis through the enhancement of HIF-1α, C-X-C chemokine receptor type 4 (CXCR4), C–C chemokine receptor type 2 (CCR2), VEGF, MMP-2 and MMP-9 factors [214]. HIF-1 decreases the expression of adiponectin as well [65].

Moreover, adiponectin may promote tumor progression through enhanced angiogenesis [224]. Research in colorectal and breast cancer shows that adiponectin has a prominent pro-angiogenic activity and that increased AdipoRs expression is associated with cancer invasiveness and/or progression [154,203,260]. Adiponectin downregulates STAT3 phosphorylation and activation, which increases tumor cell activity [36]. Complete adiponectin deficiency suppresses mammary carcinogenesis, accompanied by decreased tumor angiogenesis [54]. Lack of adiponectin noticeably reduced tumor growth and primary tumor-induced vascularization, along with increased hypoxia and apoptosis, suggesting that adiponectin might be a pro-angiogenic regulator [54,75].

T-cadherin may also be a mediator of adiponectin’s effects on tumor neovascularization. Certainly, T-cadherin expressed in the tumor vasculature promotes cancer as a pro-angiogenic factor in cooperation with adiponectin [203]. Both adiponectin and T-cadherin are colocalized in tumor vasculature, for example, in the intratumoral capillaries of human hepatocellular carcinoma (HCC) and mammary tumors, which suggests that this receptor seems to be necessary for adiponectin-mediated signaling at the level of endothelial cells [36,54,230].

CXCL1 is an angiogenesis supporter and promoter of the secretion of VEGF [261]. Adiponectin plays a pro-angiogenic role in ovarian cancer via stimulation of CXCL1 secretion from ovarian cancer cells, which promotes angiogenesis independently of VEGF. Moreover, AdipoR1, expressed in ovarian cancer cell lines, participates in new blood vessel formation as a precursor of angiogenesis [118,224]. Adiponectin stimulation of adaptor proteins containing pleckstrin homology domains, phosphotyrosine binding domain and leucine zipper motif 1 (APPL1) causes AKT-dependent phosphorylation and activation of nitric oxide synthase and then VEGF expression in endothelial cells [8]. Interaction of adiponectin with AdipoRs promotes VEGF-A expression through the activation of PI3K/AKT/m-TOR/HIF-1α signaling in human chondrosarcoma, while knockdown of adiponectin reduces VEGF-A expression and angiogenesis [216]. 

Ceramidase activity is defective in cells lacking both adiponectin receptor isoforms. Adiponectin binding to AdipoRs, independently of AMPK, stimulates ceramidase activity and enhances ceramide catabolism and the formation of its metabolite, S1P [225]. S1P is a pro-angiogenic factor and an inducer of vascular maturation, playing important roles in vasculogenesis and angiogenesis [262,263]. Moreover, S1P is involved in tumor angiogenesis [264,265].

TNF-α induces the expression of vascular cell adhesion molecule-1 (VCAM-1), which has pro-angiogenic potential and is tightly associated with tumor angiogenesis and tumor cell invasion [266]. Factors that increase VCAM expression also include leptin and adiponectin [219,220], which demonstrated another indirect manner of inducing angiogenesis by adiponectin.

On angiogenesis and vasculogenic mimicry, adiponectin has an effect mediated by COX-2. Adiponectin increases the mRNA expression and protein levels of COX-2 and activates sphingosine kinase-1 (SphK-1)/COX-2 [56] and PPARα/COX-2 signaling [228]. In pancreatic ductal adenocarcinoma (PDAC), elevated COX-2 expression promotes angiogenesis through EGFR/p38-MAPK/specificity protein-1 (Sp1)-dependent signaling [267], and inhibition of COX-2 leads to decreased angiogenesis and tumor growth in a VEGF-dependent manner [267,268].

Adiponectin enhances the production of proangiogenic MMPs such as MMP-1, MMP-2, MMP-9 and MMP-14 [37,93,125,126,127] and reduces TIMP1 activity [42,129]. Via activation of AdipoR1 and then AMPK-AKT pathways, adiponectin increases endothelial cell proliferation, migration and angiogenesis by increasing the expression of pro-angiogenic factors like VEGF, MMP-2 and MMP-9 [125]. As noted, in A549 cell culture, a NSCLC model, transfection with adiponectin successfully increased the expression levels of MMP-1, MMP-2, MMP-9 and MMP-14, providing evidence of an adiponectin-MMPs-angiogenesis-involved mechanism in NSCLC invasion [93].

On the other hand, adiponectin shows noticeable activity in preventing the growth of new blood vessels [207]. By modulating the AMPK pathway, which is the main adiponectin signaling pathway, and blocking the mTOR pathway by AMPK, adiponectin acts as a direct endogenous inhibitor of angiogenesis [59]. This adipokine decreases the survival and proliferation of several cell types, including endothelial cells [11]. It inhibits the proliferation and tube formation as well as decreases the migration of endothelial cells (HUVEC), which is also VEGF-induced and expresses AdipoRs [190]. Through decreased VEGF and anti-apoptotic Bcl-2 expression with increased pro-apoptotic p53, Bax and caspase activation, adiponectin stimulates endothelial cell apoptosis [94] as well as inhibits angiogenesis [11]. With anti-angiogenic potential, adiponectin is able to inhibit basal tube formation in macrovascular and microvascular endothelial cells while reducing VEGF-mediated migration and proliferation of these cells, with an effect comparable to that of the anti-VEGF agent—bevacizumab [202].

The anti-angiogenic function of adiponectin through induced-endothelial-cell apoptosis by activation of the p53-caspase pathway can also reduce tumor development [14]. Activation of AMPK in endothelial cells, which is the principal signaling pathway for adiponectin tumor growth suppression, inhibits tube formation caused by bone morphogenetic protein 9 (BMP9) [205]. It is proven that adiponectin, apart from suppressing angiogenesis by inducing endothelial cell apoptosis, may exert anti-neoplastic activity through inhibition of tumor proliferation and induction of apoptosis [95,207].

Adiponectin interacts with pro-angiogenic factors: IL-6 and TNF-α [97,269]. Those angiogenesis promoters reducing the expression of PPARγ decrease the expression of adiponectin [39,40,208]. However, adiponectin may also inhibit the production of TNF-α and IL-6 from TAM via the NF-κβ/cAMP-dependent pathway to reduce TNF-α-induced effects on cell proliferation and migration and regulate angiogenesis indirectly [11,40,189].

Defective tumor growth appeared to be associated with decreased neovascularization, leading to significantly increased tumor cell apoptosis. Adiponectin, as a negative modulator of angiogenesis, may exert its anti-neoplastic effects on cancer by influencing tumor angiogenesis by inducing endothelial cell apoptosis. The anti-endothelial mechanisms involve the activation of caspases 3, 8 and 9 [207,208]. Adiponectin treatment inhibits the growth of murine fibrosarcoma cells, and the underlying mechanism implicated impaired tumor growth associated with inhibition of endothelial cell proliferation and migration (inducing caspase 8-mediated apoptosis) with decreased neovascularization. Adiponectin reduces neovascularization in peritoneal metastases of gastric cancer cells as well [54]. Also in the area of angiogenesis, adiponectin and its effects on other components of TME show anti-angiogenic effects [75]. AdipoRon, which is the same as adiponectin, may also suppress the infiltration of TAMs that are known to accelerate angiogenesis [270]. VEGF and MMP-9 promote angiogenesis directly. Adiponectin treatment causes suppression of tumor angiogenesis in liver cancer cells since it downregulates ROCK/IP10/angiopoietin 1/MMP-9/VEGF cell signaling in tumor tissue, which contributes to inhibition of tube formation of tumor endothelial cells, their damage, and decreased microvessel density. In addition, the phenomenon of TAM inhibition by adiponectin synergizes with the downregulation of VEGF and MMP-9 expression in tumor tissue [95]. Treatment of human RCC cell lines with adiponectin inhibits two essential steps in the metastatic process: the secretion of VEGF and the MMP-2- and MMP-9-dependent invasion and migration, leading to a decrease in the angiogenic capacity of human RCC cells [94].

An impaired adiponectin AdipoR1-dependent mechanism mediated by PPAR-γ/PPAR-γ co-activator 1 alpha (PGC-1α) signaling inactivation contributes to a negative effect on the vasculature under conditions of ischemic stress [238]. Adiponectin is significantly downregulated in hypoxic conditions [271] and inadequate activation of the mTOR pathway leads to constitutive HIF-1 transcription, which results in angiogenesis via the induction of VEGF [94]. Thus, beyond angiogenesis, hypoxia induced by hypoadiponectinemia fosters tumors to assume a highly aggressive phenotype [54].

Human mammary BCC lines, MCF7 and MDA-MB-231, treated with adiponectin inhibited VEGF secretion [190]. In prostate cancer, adiponectin inhibits tumor cell growth by suppressing VEGF-A-mediated tumor neovascularization via AMPK/tuberous sclerosis complex 2 (TSC2), resulting in inhibition of mTOR-mediated VEGF-A activation and decreased VEGF-A production [217]. Further, miR-323 may increase VEGF-A-mediated cancer neoangiogenesis in prostate cancer cells through AdipoR1 suppression. miR-323, by binding to the 3′ untranslated regions (3′UTR) of AdipoR1 mRNA, inhibits its translation, with their levels being inversely correlated [218]. In colon cancer cells, adiponectin decreases angiogenesis by reducing the expression of angiogenic factors (CD31, VEGFb and VEGFd) and increasing the anti-angiogenesis cytokine (IL-12) [215]. AdipoRon-mediated reduction in microvessel density in tumors inhibits angiogenesis and induces necroptosis of pancreatic cancer cells [270], and likewise, a notable decrease in vessel density occurs in NPC tumors [272]. 

## 8. Other Adipokines and Carcinogenesis

The literature review revealed that most studies were performed to assess the involvement of leptin and adiponectin in carcinogenesis. However, there is evidence for the implication of other adipokines such as resistin, visfatin, apelin and chemerin in the development of cancer [19,22].

Resistin, with its proinflammatory and angiogenic properties, promotes EMT while taking part in the pathophysiological progression of breast cancer, ovarian cancer, endometrial cancer, esophageal squamous cell carcinoma, colon cancer and pancreatic cancer, which is correlated with diagnosis and prognosis [7,17,52,118,250].

Another pro-inflammatory adipokine, visfatin, like resistin, induces EMT in cancer and promotes the proliferation of different types of tumor cells [60]. Its excessive secretion is associated with a poorer prognosis for breast cancer and colorectal cancer [60,65]. Moreover, the visfatin inhibitor-FK866 in combination with cisplatin represents a promising therapeutic target for cholangiocarcinoma [273].

Apelin demonstrates regulatory effects in gastrointestinal cancers [250]. The apelin expression correlates with poor overall survival (OS). By participating in the formation of metastases, apelin is involved in the development and advanced stage of many cancers, especially breast, ovarian, RCC and gliomas [118,274].

Chemerin in cancer plays pro- and anti-tumor roles and has the potential to be a useful diagnostic and prognostic biomarker for cancer patients. Higher levels of chemerin are associated with colorectal cancer and ovarian cancer risk [52,118]. Patients with poorly differentiated prostate cancer had higher chemerin concentrations, which were associated with the Gleason score [275]. Measurement of chemerin levels may have prognostic significance in NSCLC [276].

Bariatric surgery is the most effective treatment for excess subcutaneous and visceral adipose tissue in obesity. Today, Roux-en-Y gastric bypass (RYGB), sleeve gastrectomy (SG) and adjustable gastric banding are the most popular and commonly performed bariatric surgeries (BS). However, laparoscopic sleeve gastrectomy is considered less technically challenging than laparoscopic RYGB because it does not require a gastrointestinal anastomosis or intestinal bypass [277]. Studies have shown that bariatric surgery may change adipokine levels, therefore indirectly affecting the course of cancer. According to the literature, bariatric surgery is associated with a reduced overall incidence of cancer (RR 0.62, 95% CI 0.46–0.84, *p* < 0.002), obesity-related cancer (RR 0.59, 95% CI 0.39–0.90, *p* = 0.01) and cancer-associated mortality (RR 0.51, 95% CI 0.42–0.62, *p* < 0.00001) [278].

## 9. Conclusions

This review demonstrates that changing levels of leptin and adiponectin have a significant impact on the incidence, progression and reoccurrence of many cancers; therefore, a correct proportion and relationship between them, which describes the ratio of leptin to adiponectin [8,273], are crucial.

The overwhelming evidence on the pivotal role of these adipocytokines in cancer allows us to infer that they may be used in diagnostics, which could serve as prognostic and predictive biomarkers for many types of cancer [279]. Leptin concentrations and/or the expression of ObRs in tumors could be used as potential tumor markers in the diagnosis and prognosis of cancer [12]. The higher levels of leptin and/or ObRs in patients with tumor invasion and distant metastasis [4] and their correlation with decreased relapse-free survival [8] support the consideration of leptin as a poor prognostic factor. Moreover, leptin may predict the response to a therapeutic intervention because it affects the response to treatment and its effectiveness [190]. Patients with higher levels of leptin mRNA expression are less responsive to treatment, while those with low levels are more likely to survive [6]. Due to their significant inverse correlation with the number of tumors, adiponectin and AdipoRs may be considered prognostic factors [93,280]. Low adiponectin is connected to poor prognosis and possibly carcinogenesis, as well as decreased AdipoRs expression associated with a histologically higher grade of cancer [6].

Determining circulating levels of adipokines, the discovery of their receptors in various cancers and the recognition of their downstream signaling pathways in the context of cancer are areas of research that could bring novel therapeutic targets for managing adipocytokines for the prevention and treatment of human cancers [1,8,19]. Counteracting or inhibiting the leptin proliferation signaling pathway may represent a potential therapeutic target [9,13,64,281]. Therapeutic strategies to increase adiponectin concentrations may serve as a therapeutic tool for tumor patients [14,19,75] and treatment with adiponectin may modulate the poor prognosis of cancer patients [6].

Due to their real promise in improving cancer diagnostics and more effective therapeutic strategies, understanding the correlations and roles of those adipokines in cancer. As well as an exact dissection of the underlying mechanism by which these disrupted adipokines promote progression and metastasis, further clarification with more research and longitudinal studies is requsired.

## Figures and Tables

**Figure 1 cancers-15-04250-f001:**
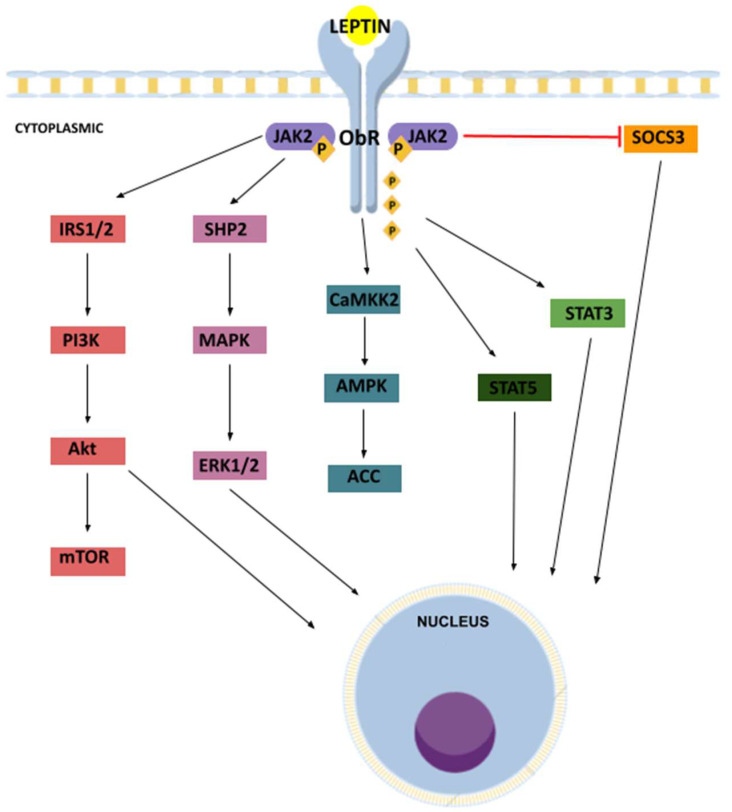
Leptin and a schematic representation of leptin-induced signaling pathways. The binding of leptin to its receptor leads to the formation of the ObR/JAK2 complex, which results in phosphorylation (P) and activation of STAT3 and STAT5, which are translocated to the nucleus and activate transcription of target genes, including the gene for SOCS3. Chronic stimulation leads to an increase in SOCS3, which negatively regulates leptin signaling by inhibiting JAK2 activity. JAK2 phosphorylation also leads to activation of SHP2, leading to increased MAPK/ERK1/2 signaling and phosphorylation of IRS1/2, which recruits PI3K to activate downstream signals. mTOR is an important downstream target of PI3K/Akt in the leptin signaling pathway, promoting cell growth and survival. In addition, leptin regulates metabolism through AMPK/ACC signaling in the brain and peripheral organs. AMPK activation may occur via a STAT3-independent signaling pathway. Blocking AMPK activation inhibits the phosphorylation of ACC stimulated by leptin. ACC, acetyl-CoA carboxylase; Akt, protein kinase B; AMPK, 5′-AMP-activated protein kinase; CaMKK2, calcium/calmodulin-dependent protein kinase; ERK, extracellular-signal-regulated kinase; IRS, insulin receptor substrates; JAK2, Janus kinase 2; MAPK, mitogen activated protein kinase; mTOR, the mammalian target of rapamycin; ObR, leptin receptor; PI3K, phosphatidylinositol 3-kinase; SHP2, Src Homology 2 domain; SOCS3, suppressors of cytokine signaling 3; STAT3, activator of transcription 3; STAT5, activator of transcription 5. Black arrows indicate activation of the target protein, whereas a small perpendicular red line at the end of the red lines indicates inhibitory effects. The figure was created by mindthegraph.com (accessed on 6 March 2023).

**Figure 2 cancers-15-04250-f002:**
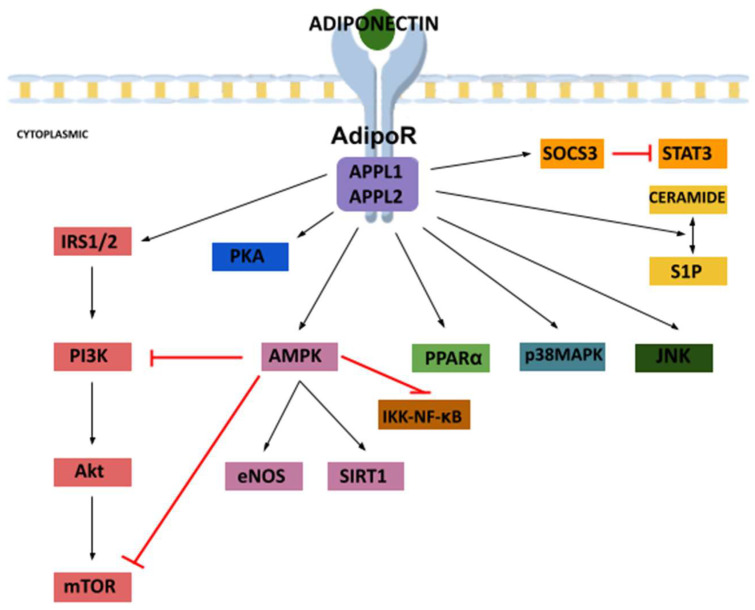
Adiponectin and a schematic representation of adiponectin-induced signaling pathways. The binding of adiponectin to its receptor leads to the recruitment of APPL1 and APPL2, thereby activating a number of downstream signaling pathways. Adiponectin effects are mostly mediated via the AMPK and PPARα pathways. Stimulation of AMPK results in activation of SIRT1, essential in adiponectin’s regulation of glucose and lipid homeostasis, as well as enhanced eNOS activity, through which adiponectin interacts with endothelial cells. Moreover, activation of AMPK suppresses PI3K, mTOR and IKK/NF-κB signaling, exerting a cytoprotective effect of adiponectin. Adiponectin binding to AdipoRs, independently of AMPK, stimulates ceramidase activity and enhances ceramide catabolism and the formation of its metabolite—S1P, which is involved in angiogenesis. Activation of IRS1/2 by adiponectin causes increased PI3K/Akt/mTOR signaling, controlling cell survival, growth and apoptosis. Adiponectin signaling also activates other downstream intracellular signaling cascades through PKA, JNK and p38MAPK. Via SOCS3, adiponectin inhibits STAT3 activation, which increases proliferation, survival and invasion of cancer cells and suppresses anti-tumor immunity. ACC, acetyl-CoA carboxylase; AdipoR, adiponectin receptor; Akt, protein kinase B; AMPK, 5′-AMP-activated protein kinase; APPL1, adaptor protein containing a pleckstrin homology domain 1 protein; APPL2, adaptor protein containing a pleckstrin homology domain 2 protein; eNOS, endothelial nitric oxide synthase; IKK, IκB kinase; IRS, insulin receptor substrates; JNK, -Jun N-terminal kinase; MAPK, mitogen activated protein kinase; mTOR, the mammalian target of rapamycin; NF-κB, nuclear factor-κB; PI3K, phosphatidylinositol 3-kinase; PPARα, peroxisome proliferator-activated receptor gamma; S1P, *sphingosine-1-phosphate*; SIRT1, sirtuin 1; SOCS3, suppressors of cytokine signaling 3; STAT3, activator of transcription 3. Black arrows indicate activation of target protein, whereas small perpendicular red lines at the ends of the red lines indicate inhibitory effects. The figure was created by mindthegraph.com (accessed on 6 March 2023).

**Figure 3 cancers-15-04250-f003:**
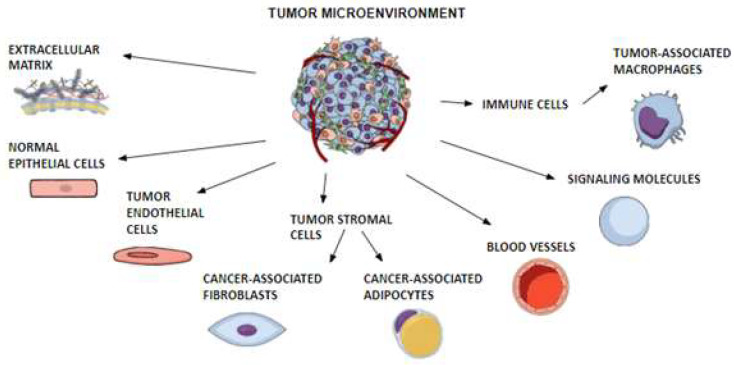
Composition of the tumor microenvironment. A schematic diagram shows the different components of the tumor microenvironment. The dynamic and bidirectional interactions of tumor cells with their microenvironment, consisting of cellular and non-cellular parts, are fundamental to the stimulation of tumor growth, invasion and metastasis. The figure was created by mindthegraph.com (accessed on 6 March 2023).

**Table 1 cancers-15-04250-t001:** Leptin and adiponectin interactions with tumor microenvironment.

Component	Adipokine	Cancer Types	Effect	Mechanisms	References
TME	↑ Leptin				[64]
	↓ Adiponectin				[69]
CAFs	Leptin	Breast cancer	↑ Proliferation ↑ Migration ↑ Invasion		[9,72]
		NSCLC	↑ Malignancy		[73]
		Pancreatic cancer	↑ Invasion		[74]
	Adiponectin	Colon cancer	↑ Angiogenesis ↑ Tumor growth ↑ Proliferation ↑ Migration ↑ Invasion		[75] [76]
TAMs	Leptin	Breast cancer	↑ Malignancy ↑ Tumor growth ↑ Progression		[77,78,79]
		Melanoma	↑ Metastasis		[80]
		Gallbladder cancer	↑ Invasion ↑ Migration		[81]
		Colorectal cancer	↑ Tumor growth		[82]
	Adiponectin	Melanoma	↓ Tumor growth		[83]
		Lung cancer	↓ Tumor growth		[83]
		Rhabdomyosarcoma	↓ Tumor growth		[84]
MMPs	Leptin	Gastric cancer	↑ Invasion ↑ Metastasis	MMP-1	[85]
		Breast cancer	↑ Progression	MMP-2 MMP-9	[64,86,87]
		Oesophageal cancer	↑ Invasion	MMP-2 MMP-9	[88]
		Gallbladder cancer	↑ Metastasis	MMP-3 MMP-9	[89]
		Ovarian cancer	↑ Migration ↑ Invasion	MMP-7	[90]
		Colon cancer	↑ Progression	MMP-7	[91]
		Pancreatic cancer	↑ Migration ↑ Invasion	MMP-13	[92]
	Adiponectin	NSCLC	↑ Invasion	MMP-1 MMP-2 MMP-9 MMP-14	[93]
		RCC	↓ Tumor growth ↓ Metastasis ↓ Angiogenesis	MMP-2 MMP-9	[94]
		Liver cancer	↓ Tumor growth ↓ Metastasis	MMP-9	[95]
		Oesophageal cancer	↓ Invasion	MMP-2 MMP-9	[88]

Abbreviations: CAFs, cancer-associated fibroblasts; MMPs, matrix metalloproteinases; NSCLC, non-small-cell lung cancer; RCC, renal cell carcinoma; TAMs, tumor-associated macrophages; TME, tumor microenvironment.

**Table 2 cancers-15-04250-t002:** The effect of leptin and adiponectin on angiogenesis.

Angiogenic Process or Factor	Adipokine	Mechanisms or Involved Intracellular Signaling Pathways	Reference
Proliferation, migration and differentiation of endothelial cells	Leptin	JAK2/STAT3 JAK2/STAT3/SOCS3 NF-κB ERK1/2 ERK2 Akt Wnt PI3K COX-2 CACs, Src kinase and integrin αvβ5	[4,6,12,190,191,192,193] [194] [195,196] [195] [12] [193,197] [197] [12] [12] [198]
	Adiponectin	STAT3 AMPK Akt Ras/ERK1/2 MAPK eNOS/NO PI3K The cascade activation of caspase-8, -9 and -3	[11,154,190,199,200,201,202] [36] [54,59,75,125,189,203,204,205,206] [8,75,125,189,203] [203] [189,203] [8,189,203,204] [75] [54,207,208]
VEGF	Leptin	MAPK, PI3K JNK, p38 MAPK, PKC PI3K/AKT/mTOR/S6 kinase MAPK (p38, ERK and JNK) miR-122/PKM2 Akt and Wnt mTOR JAK2/PI3K/ERK/mTOR	[6,10,190,191,192,197,209,210] [196] [196] [211] [212] [156] [197] [213] [12]
	Adiponectin	AMPK-Akt ROCK/IP10/angiopoietin 1/MMP-9/VEGF PI3K/Akt/m-TOR/HIF-1α AMPK/TSC2 miR-323	[75,214,215] [94,125,190] [95] [216] [217] [218]
FGF	Leptin		[10,191,192]
	Adiponectin		[6]
PDGF	Leptin	Akt and Wnt	[197]
LIF	Leptin		[12]
VCAM	Leptin		[219]
	Adiponectin	NF-kB/COX-2	[219,220]
IL-1	Leptin	JAK2/STAT3 MAPK/ERK1/2 PI3K/Akt1 PKC p38 JNK NF-kB	[221,222,223] [221,222,223] [221,222,223] [221] [221,222,223] [221,222,223] [221,222,223]
IL-1β	Leptin	JAK2/PI3K/ERK/mTOR	[12]
IL-6	Leptin		[10]
	Adiponectin	NF-κB/cAMP	[11,189]
IL-12	Adiponectin		[215]
CD31	Leptin	Akt and Wnt	[154] [197]
	Adiponectin		[215]
CD144	Leptin	Akt and Wnt	[197]
CXCL1	Adiponectin		[75,214,224]
CCR2	Adiponectin		[214]
CXCR4	Adiponectin		[214]
S1P	Adiponectin	Ceramidase activity	[54,225]
COX-2	Leptin	p38MAPK/Akt/COX-2	[226] [227]
	Adiponectin	SphK-1/COX-2 NF-kB/COX-2 PPARα/COX-2	[56] [219,220] [228]
TF	Leptin		[229]
T-cadherin	Adiponectin		[36,54,199,203,214,230]
ROS	Leptin		[231,232]
Hypoxia	Leptin	HIF-1α	[233,234,235,236,237] [156,210]
	Adiponectin	mTOR/HIF-1 PPARγ/PGC-1α	[54,214] [94] [238]
MMP-1	Leptin		[85]
	Adiponectin		[93,127]
MMP-2	Leptin		[64,86,87,88,173,239]
	Adiponectin		[93,94,125,127,214]
MMP-7	Leptin		[90,91]
MMP-9	Leptin		[64,86,87,88,173,239]
	Adiponectin	ROCK/IP10/angiopoietin	[75,93,94,125,126,127,214] [95]
MMP-14	Leptin		[120]
	Adiponectin		[93]
TIMP1	Adiponectin		[42,129]
EMT	Leptin		[240]

Abbreviations: Akt, protein kinase B; AMPK, 5′-AMP-activated protein kinase; CACs, circulating angiogenic cells; cAMP, cyclic adenosine 3′,5′-monophosphate; CCR2, C–C chemokine receptor type 2; COX-2, cyclooxygenase-2; CXCL1, CXC chemokine ligand 1; CXCR4, chemokine receptor type 4; EMT, the epithelial–mesenchymal transition; eNOS, endothelial nitric oxide synthase; ERK, extracellular-signal-regulated kinase; FGF, fibroblast growth factor; HIF, hypoxia-inducible factor; IP10, IFN-inducible protein 10; JAK2, Janus kinase 2; JNK, c-Jun N-terminal kinase; LIF, leukemia inhibitory factor; MAPK, mitogen activated protein kinase; miR, microRNA; MMP, matrix metalloproteinase; mTOR, the mammalian target of rapamycin; NF-κB, nuclear factor-κB; NO, nitric oxide; PDGF, platelet-derived growth factor; PGC-1α, PPAR-γ co-activator 1 alpha; PI3K, phosphatidylinositol 3-kinase; PKC, protein kinase C; PKM2, pyruvate kinase M2; PPARα, peroxisome proliferator-activated receptor alpha; PPARγ, peroxisome proliferator-activated receptor gamma; Ras, rous sarcoma kinase; ROCK, Rho-associated protein kinase; ROS, reactive oxygen species; S1P, sphingosine-1-phosphate; SOCS3, suppressors of cytokine signaling 3; SphK-1, sphingosine kinase-1; Src, steroid receptor coactivator; STAT3, activator of transcription 3; TIMP1, metalloproteinase inhibitor 1; TF, tissue factor; TSC2, tuberous sclerosis complex 2; VCAM, vascular cell adhesion molecule; VEGF, vascular endothelial growth factor. Wnt, wingless-related integration site.

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
