# Peer review of "Role of Leptin and Adiponectin in Carcinogenesis"

_cancers, 2023, doi:10.3390/cancers15174250_

Round 1
Reviewer 1 Report
I was glad to review the work of the authors regarding this very interesting review on the Role of leptin and adiponectin in carcinogenesis. The manuscript is well-written and the incorporated tables and figures make the study easy to follow.
I strongly recommend acceptance for publication of the paper after minor changes.
1) I would suggest a brief discussion on other adipokines as well
2) "Today, Roux-en-Y gastric bypass (RYGB), sleeve gastrectomy (SG), and adjustable gastric banding are the most popular and commonly performed bariatric surgeries (BS). However, Laparoscopic sleeve gastrectomy is considered less technically challenging than laparoscopic RYGB because it does not require a gastrointestinal anastomosis or intestinal bypass" [1]
According to the literature, bariatric surgery is associated with a reduced overall incidence of cancer (RR 0.62, 95% CI 0.46-0.84, p < 0.002), obesity-related cancer (RR 0.59, 95% CI 0.39-0.90, p = 0.01) and cancer-associated mortality (RR 0.51, 95% CI 0.42-0.62, p < 0.00001) [2].
I would suggest adding this information in the discussion section and consider citing the articles:
1) https://pubmed.ncbi.nlm.nih.gov/35069068/
2) https://pubmed.ncbi.nlm.nih.gov/37047163
General comments
The spelling and punctuation are very good. No issues were detected.
Abstract
The abstract is concise. All the necessary information about the study is included.
Background
- The information provided in the introduction is important for the comprehension of the article.
- The objective of the study is clearly mentioned.
Methods
- The methods are sufficiently explained by the authors.
Results
- The results are presented in a very extensive way.
- The table is really helpful and necessary for the completion of the authors' work.
Discussion
- The discussion is of great quality and includes updated data.
- The authors inform the reader about the study's limitations.
Conclusion
From the presented data, the conclusion is complete and represents the work that the authors did.
Reviewer 2 Report
The aim of this review is to clarify the various mechanisms which link leptin, adiponectin and carcinogenesis for the future potential use of these adipokines in cancer diagnostics and therapeutics.
This is an interesting review of the relationship of leptin and adiponectin with the presence of cancer. I allow me to make two observations:
Please, review figure 1, as the lines for the PI3K, MAPK and MAPK pathways should be connected to the nucleo?
It would be informative if they include a figure that summarizes the involvement of both adipokines in cancer (like a graphical abstract).
Minor revision of the English
Reviewer 3 Report
The present manuscript entitled “Role of leptin and adiponectin in carcinogenesis” outlines an important contribution to the literature in the field of obesity and cancer. There have been many research papers on the role of leptin and adiponectin in the progression of cancer. Therefore, to improve the quality of this article, authors should emphasize the new findings and novelties of this manuscript. Overall, the paper is thoughtful and generally supports the conclusions of the authors. After addressing the following comments the current manuscript has significance for the publication in this esteemed journal.
1. The figures are easy to understand. However, I suggest not to crop the nucleus drawing in Figure 1, because in the present form it seems to be cut away from the image.
2. To make the meaning of Figure 1 and Figure 2 clearer, I would add a black arrow at the end of the black lines (that indicate the activation) and a small perpendicular red line to the end of the red lines (to indicate the inhibitory effect). In this way, the meaning of the interactions described in the figure would be more intuitive.
3. I furthermore suggest to increase the resolution of all the figures.
4. Some concepts that recur in different parts of the manuscript seem to be redundant, therefore I suggest to cut the repetitive sentences.
5. I lastly suggest to check spelling and English language throughout the manuscript
I suggest to perform an English editing and a general proofreading of the manuscript.
